# Fatal *Hymenoptera* Venom–Triggered Anaphylaxis in Patients with Unrecognized Clonal Mast Cell Disorder—Is Mastocytosis to Blame?

**DOI:** 10.3390/ijms242216368

**Published:** 2023-11-15

**Authors:** Matija Rijavec, Jezerka Inkret, Urška Bidovec-Stojković, Tanja Carli, Nina Frelih, Andreja Kukec, Peter Korošec, Mitja Košnik

**Affiliations:** 1University Clinic of Respiratory and Allergic Diseases Golnik, 4204 Golnik, Slovenia; urska.bidovec-stojkovic@klinika-golnik.si (U.B.-S.);; 2Biotechnical Faculty, University of Ljubljana, 1000 Ljubljana, Slovenia; 3Institute of Forensic Medicine, Faculty of Medicine, University of Ljubljana, 1000 Ljubljana, Slovenia; 4National Institute of Public Health, 1000 Ljubljana, Slovenia; 5Faculty of Medicine, University of Ljubljana, 1000 Ljubljana, Slovenia; 6Faculty of Pharmacy, University of Ljubljana, 1000 Ljubljana, Slovenia; 7Faculty of Medicine, University of Maribor, 2000 Maribor, Slovenia

**Keywords:** fatal anaphylaxis, *Hymenoptera* venom, mastocytosis, venom immunotherapy

## Abstract

*Hymenoptera* venom–triggered anaphylaxis (HVA) affects up to 8.9% of the general population and is the most frequent cause of anaphylaxis in adults, accounting for approximately 20% of all fatal anaphylaxis cases. Quite often, a fatal reaction is a victim’s first manifestation of HVA. Mastocytosis represents one of the most important risk factors for severe HVA. We analyzed patients with documented fatal HVA for the presence of underlying clonal mast cell disorder (cMCD). Here, we report three cases of fatal HVA, with undiagnosed underlying cMCD identified by the presence of the peripheral blood and/or bone marrow *KIT* p.D816V missense variant postmortem. In the first case, anaphylaxis was the initial episode and was fatal. In the other two cases, both patients were treated with specific venom immunotherapy (VIT), nevertheless, one died of HVA after VIT discontinuation, and the other during VIT; both patients had cardiovascular comorbidities and were taking beta-blockers and/or ACE inhibitors. Our results point to the importance of screening all high-risk individuals for underlying cMCD using highly sensitive molecular methods for peripheral blood *KIT* p.D816V variant detection, including severe HVA and possibly beekeepers, for proper management and the need for lifelong VIT to prevent unnecessary deaths. Patients at the highest risk of fatal HVA, with concomitant cardiovascular and cMCD comorbidities, might not be protected from field stings even during regular VIT. Therefore, two adrenaline autoinjectors and lifelong VIT, and possibly cotreatment with omalizumab, should be considered for high-risk patients to prevent fatal HVA episodes.

## 1. Introduction

Anaphylaxis is an acute, severe, systemic, and life-threatening hypersensitivity reaction that requires immediate medical attention [1,2]. Exposure to various allergens leads to the activation of mast cells and basophils. The reaction is mostly triggered by food, drugs, and insect stings, and without treatment, anaphylaxis can be fatal due to its rapid progression to respiratory and/or cardiac arrest/collapse [1,2].

*Hymenoptera* venom–triggered anaphylaxis (HVA) is relatively common since approximately 56–94% of the adult population is estimated to be stung by a *Hymenoptera* at least once in a lifetime, with 0.3 to 8.9% presenting systemic reactions [3,4,5]. Although the exact incidence of fatal anaphylaxis remains challenging to estimate accurately due to underreporting and misdiagnosis, according to European Anaphylaxis Registry data [6], HVA is the most frequent cause of anaphylaxis in adults (48%) and responsible for approximately 20% of all fatal anaphylaxis cases. A fatal reaction is often the first manifestation of HVA in a victim [4]. Estimated HVA-related mortality varies between 0.03 and 0.48 per million population per year, consequently it is among the top three causes of anaphylaxis-related deaths worldwide [3,4,7,8,9]. The frequency of *Hymenoptera* stings and subsequent risks for developing allergic reactions are highly influenced by geographic, environmental, and ecological factors, hence they vary substantially between different countries/regions [3,4,7,8,9].

Clonal mast cell diseases (cMCD), such as mastocytosis and monoclonal mast cell activation syndrome, represent some of the most important long-term risk factors for severe HVA, with severe HVA the presenting symptom in cMCD in up to 50% of cases. The association between cMCD and severe HVA has been known for years. However, recent peripheral blood screening of the *KIT* p.D816V missense variant revealed the true extent of this clinical association. An activating somatic missense variant in the *KIT* gene, p.D816V (c.2447A > T), which promotes mast cell proliferation observed in cMCD, is present in the vast majority of patients with mastocytosis. The presence of this variant represents one of the minor criteria for the diagnosis of systemic mastocytosis (SM) [10]. The proposed role of the *KIT* p.D816V variant and cMCD in fatal HVA is presented in Figure 1.

In two recent large studies, we found cMCD (determined by the presence of *KIT* p.D816V in peripheral blood) among 21% of patients with severe (Mueller grade IV) HVA [3,11,12]. In contrast to classic SM, where elevated basal serum tryptase (BST) represents a diagnostic criterion, the vast majority (82%) of patients with HVA and cMCD have been found to show normal tryptase levels and would have been missed if BST levels were used as a gating criterion for SM (cMCD) evaluation [3,10,11,12]. Hence, as recently proposed [13], peripheral blood screening for the *KIT* p.D816V variant should be performed in all high-risk HVA patients regardless of BST levels. HVA, particularly in the context of mastocytosis, represents a significant clinical concern. Patients with cMCD and HVA represent a particular risk group for very severe anaphylactic reactions, and several fatal HVA cases have been reported in such patients [14,15,16,17]. Therefore, identifying patients with HVA and cMCD is of utmost importance for the effective prevention and management of such patients. All patients with concomitant HVA and cMCD should be equipped with two adrenaline autoinjectors, and lifelong venom immunotherapy (VIT) should be performed as these patients appear to have higher relapse rates after VIT [3,18].

Here, we report three cases of fatal HVA, with undiagnosed underlying cMCD identified by the presence of the peripheral blood and/or bone marrow *KIT* p.D816V missense variant postmortem.

## 2. Results

### 2.1. Fatal Anaphylaxis in Slovenia

According to the National Register of Causes of Death database, there were 27 documented fatal anaphylaxis cases over an observed 10-year period. The average annual number of deaths related to anaphylaxis was 2.7, corresponding to a mortality rate of 1.28 per million population per year. In 15 people (mean age 55.6 years, 60% were male), an autopsy was conducted at the University of Ljubljana, Faculty of Medicine, Institute of Forensic Medicine, and confirmed fatal anaphylaxis, which in 7 people (47%) was caused by *Hymenoptera* stings (in 4 patients, yellow jacket), followed by drugs in 3 people (20%), and tear gas in 1 person (7%), while in 4 people (27%), the trigger could not be identified. This resulted in a minimal definite mortality rate for anaphylaxis related to insect venom of 0.33 per million population per year. Insect venom was Slovenia’s most common trigger of fatal anaphylaxis in documented pathology-confirmed cases.

In a period of 1 year, we documented three additional cases of fatal HVA in Slovenia, which are described in detail.

### 2.2. Fatal HVA in a Beekeeper (Case 1)

A 71-year-old man with a history of coronary disease and arterial hypertension was using beta-adrenergic blockers and angiotensin-converting enzyme inhibitors to treat his condition. He had no allergy history, was a beekeeper for over 50 years, and tolerated several hundred bee stings yearly. After being simultaneously stung by over 100 bees, mostly in his chest, he vomited and was found unconscious. Upon arrival at the emergency medical service, he was without cardiac activity, which was temporarily restored after defibrillation; he received several doses of epinephrine. During helicopter transport, electromechanical dissociation occurred.

The autopsy revealed at least 100 bee stings, severe stenosis of the coronary arteries and ischemic changes in myocytes. The postmortem serum tryptase level was 1530 ng/mL, which was high above the suggested postmortem tryptase cut-off for the diagnosis of anaphylaxis that ranged from 44.3 [19] to 53.8 ng/mL [20]). His sIgE levels were negative for honeybee venom and its recombinant allergens (rApi m 1, rApi m 3, rApi m 5, and rApi m 10), yellow jacket venom and its recombinant allergens (rVes v 1 and rVes v 5), Polistes venom, and hornet venom, but was slightly positive for the honeybee recombinant allergen rApi m 2 (0.22 kIU/L). In patients with HVA and cMCD, sIgE levels of 0.10 kU/L or more are significant, as those patients often have low sIgE levels.

*KIT* p.D816V was detected postmortem in the patient’s peripheral blood and bone marrow, demonstrating undiagnosed cMCD. The patient had normal tryptase copy numbers (α,β/α,β).

### 2.3. Fatal HVA during VIT in a Patient with Normal BST (Case 2)

A 31-year-old man with a history of tachycardia was using beta-adrenergic blockers to treat his condition. The man had a known history of HVA, with an episode of severe Mueller grade IV reaction at the age of 26, involving a loss of consciousness in the absence of skin symptoms 10 min after being stung by a yellow jacket. Emergency medical services successfully rescued the patient.

The allergy work-up confirmed sensitization to the yellow jacket by an intradermal skin test (positive only for the yellow jacket and negative for the honeybee) and sIgE (positive for yellow jacket venom sIgE and no sIgE against honeybee venom). Patient-initiated VIT with yellow jacket venom extract was well tolerated. BST was normal (3.59 ng/mL). After 4 years of yellow jacket VIT, he was stung by a single yellow jacket, collapsed, and died.

The postmortem serum tryptase level was 152 ng/mL, and the patient was negative for sIgE for the honeybee, yellow jacket, Polistes, and hornet venoms, and their recombinant allergens (rApi m 1, rApi m 2, rApi m 3, rApi m 5, rApi m 10, rVes v 1, and rVes v 5). The *KIT* p.D816V missense variant was detected in the patient’s peripheral blood, again demonstrating undiagnosed cMCD. The patient had normal tryptase copy numbers (β,β/β,β).

### 2.4. Fatal HVA after VIT Discontinuation (Case 3)

A 61-year-old woman had a known history of HVA, first after being stung by a yellow jacket at the age of 50 (Mueller grade III) and then by a honeybee 3 years later (Mueller grade IV). Allergy work-up confirmed double sensitization to honeybee and yellow jacket by intradermal, sIgE, inhibition, and basophil activation tests. BST was highly elevated (35.1 ng/mL). The patient completed yellow jacket VIT, while honeybee VIT was discontinued due to severe side effects. Six years after stopping wasp venom VIT, she was stung by several yellow jackets, collapsed, and died.

We confirmed the presence of *KIT* p.D816V in peripheral blood from the archival sample, again demonstrating undiagnosed cMCD. The patient had normal tryptase copy numbers (α,β/β,β).

## 3. Discussion

Fatal HVA is a rare condition, but its social impact in terms of life expectancy or potential years of life lost is high; therefore, its prevention requires substantial medical attention. In a period of 1 year, we documented three cases of fatal HVA in Slovenia, with approximately 2.1 million inhabitants. Herein, we presented three cases of fatal HVA and highlighted that all three had undiagnosed cMCD, which was identified postmortem by the presence of the peripheral blood and/or bone marrow *KIT* p.D816V missense variant. In one case, fatal HVA occurred as the first reaction to honeybee venom.

In a recent report on HVA-related deaths in Europe, HVA-related mortality varied between 0 and 2.24 (average 0.26) per million inhabitants per year [21]. Slovenia was among the countries with the highest reported median mortality rate (0.55 fatalities per million per year) [21]. This aligned with our findings, where we reported a minimal definite pathology-confirmed HVA-related mortality rate of 0.33 per million population per year. Insect venom was the most common trigger of fatal anaphylaxis in Slovenia. This calculation considered only definite HVA-related deaths and, therefore, highly likely underestimated the exact rate of fatal anaphylaxis in Slovenia. Based on the National Register of Causes of Death database, the average annual anaphylaxis-related mortality rate over a 10-year period was 1.28 per million population per year. However, the minimal definite pathology-confirmed mortality rate for all anaphylaxis triggers was 0.71 per million population per year. Interestingly, Bilo et al. [8] reported that average definite and possible HVA-related mortality rates in Italy were 0.03 and 0.17 per million population per year, respectively, and were more common in northern Italy.

Official fatal anaphylaxis figures probably underrepresent the actual number of deaths since diagnosing fatal anaphylaxis can be challenging, especially in cases where the patient had no previous history of anaphylaxis. A detailed postmortem work-up, including serological testing for tryptase and specific IgE screening, is extremely helpful in confirming the diagnosis [19,20,22,23]. Although the stability of postmortem tryptase levels has been questioned, it is generally accepted that if femoral vein blood is taken and tests conducted within 48 h following death, tryptase levels above 44 or 54 ng/mL can support an anaphylaxis diagnosis [19,20,22,23]. Two of our patients, for whom blood was taken, had postmortem tryptase levels well beyond this cut-off, specifically 152 and 1530 ng/mL, supporting an anaphylaxis diagnosis. Unfortunately, no postmortem blood samples were available for the third patient.

Known risk factors for severe, fatal, and near-fatal anaphylaxis include male sex, older age, cardiovascular comorbidities, and mastocytosis (cMCD) [7,17,24]. Interestingly, male sex and older age are both risk factors for cMCD as well [13]. None of our patients had hereditary α-tryptasemia (HαT), a genetic trait caused by increased α-tryptase–encoding *TPSAB1* copy number, which was associated with severe HVA [3,11,25]. Although the association between cMCD and severe HVA has been known for years, recent studies employing peripheral blood *KIT* p.D816V screening have only highlighted the true extent of this clinical association and further revealed that cMCD is often present in patients with normal BST levels [11,12,26,27,28]. A recent study confirmed a high correlation between blood and bone marrow *KIT* p.D816V allele burden, demonstrating that the presence of *KIT* p.D816V in peripheral blood, which is a minor criterion for an SM diagnosis, was highly specific for cMCD, mostly (indolent) systemic mastocytosis ((I)SM), bone marrow mastocytosis (BMM) or monoclonal mast cell activation syndrome (c-MCAS) [29]. cMCD is a rare, largely underrecognized disease because it often reveals itself as secondary to another condition, typically anaphylaxis [13,17]. Therefore, it is essential to increase awareness of the severe HVA association with cMCD, especially if anaphylaxis presents without skin involvement, irrespective of BST levels [12,13,17]. Screening for and identifying patients with concomitant cMCD and HVA is of utmost importance to avoid fatal HVA, as all patients should be equipped with two adrenaline autoinjectors, and lifelong VIT as a lifesaving therapy is advised [3,13,16,18]. Severe HVA reactions have been reported after VIT discontinuation, urging the necessity of lifelong VIT in such patients [18], as highlighted by our third fatal HVA case. Furthermore, our second patient, who had a known history of tachycardia and was using beta-adrenergic blockers, died despite ongoing VIT, demonstrating that patients at the highest risk for severe HVA, i.e., those with concomitant cardiovascular comorbidities and cMCD, are not completely protected when stung in the field, even while maintaining a regular course of VIT. This is the first documented case of fatal HVA in a patient associated with cardiovascular comorbidities and mastocytosis (cMCD) during VIT. In addition to allergen avoidance and two adrenaline autoinjectors, omalizumab should be considered with VIT to prevent fatal episodes in patients with severe HVA and concomitant cardiovascular and cMCD comorbidities. Adding omalizumab to VIT in HVA patients with severe HVA and mastocytosis who do not tolerate VIT was shown to result in tolerance to VIT [30], while omalizumab alone prevented anaphylaxis in patients with mastocytosis [31], showing the usefulness of such an approach. Our case where the initial HVA episode occurred in a beekeeper as a first reaction to honeybee venom points to the importance of identifying (screening for sensitization and peripheral blood *KIT* p.D816V variant) and treating at-risk individuals, suggesting the necessity of screening high-risk groups, particularly people at high risk for multiple stings such as beekeepers. Some authors propose that VIT should be indicated in HVA-sensitized people with mastocytosis (cMCD) even without preceding HVA due to the high concomitance of cMCD and HVA [14]. This would especially apply to high-risk groups, such as beekeepers; however, more data on the rationale of such prophylactic VIT are needed. However, injecting aluminum adjuvants might generate the risk of unwanted side effects; therefore, VIT with allergen extracts without aluminum hydroxide adjuvants is recommended.

Fatal HVA in patients with cMCD has been reported previously [14,15,16,17,24,32], but the true extent of mortality rates due to fatal HVA in patients with cMCD remains unknown. A recent study analyzing Allergy-Vigilance Network data [24] reported three fatal HVA cases between 2002 and 2020, and in two patients who died, cMCD was strongly suspected given their medical history, clinical anaphylaxis presentation, acute tryptase levels, and REMA scores, but no bone marrow biopsies were performed [24]. Similarly, our report suggests that concomitant cMCD might be the most common cause of fatal HVA. In all three patients with documented cases of fatal HVA in a period of 1 year, we identified undiagnosed cMCD postmortem. That alone accounts for a mortality rate of 1.42 per million population per year.

Furthermore, we recently demonstrated that severe HVA with loss of consciousness was frequent in patients with cMCD, as more than half (56%) of patients with *KIT* p.D816V detected in the peripheral blood, indicative of cMCD, lost consciousness during HVA [12]. This further supports the observations that systemic HVA reactions are not only more common in patients with mastocytosis/cMCD but are also more severe, often life-threatening, near-fatal, and fatal [3,12,16,24]. However, to confirm the true extent of associations between fatal HVA and cMCD and to confirm whether the majority, if not all, fatal anaphylaxis cases are attributable to underlying mastocytosis/cMCD, larger, possibly multicenter studies are needed. Nevertheless, screening for underlying cMCD (by peripheral blood *KIT* p.D816V variant detection using highly sensitive molecular methods) is indicated in all patients with severe anaphylaxis, especially HVA [12,13], to avoid unnecessary deaths. Furthermore, special attention and screening for cMCD should also be performed in patients with early onset osteoporosis since the condition is a known associated cMCD risk factor. Moreover, screening for sensitization and cMCD in high-risk groups, such as beekeepers, seems plausible.

## 4. Materials and Methods

### 4.1. National Register of Causes of Death Database

A retrospective cohort study was conducted using routinely reported data from 2010 to 2019 from the National Register of Causes of Death database, managed by the National Institute of Public Health (NIPH) of the Republic of Slovenia, to identify anaphylaxis-related deaths. For the identification and in accordance with the 11th Revision of the International Statistical Classification of Diseases and Related Health Problems (ICD-11), published by the World Health Organization (WHO) [33], the following codes were used: W57 bitten or stung by nonvenomous insects and other nonvenomous arthropods; X23 contact with hornets, wasps, and bees (including yellow jackets); and X29 contact with an unspecified venomous animals or plants.

For some of our patients, autopsy data were available at the University of Ljubljana, Faculty of Medicine, Institute of Forensic Medicine.

The study was approved by the National Medical Ethics Committee of the Republic of Slovenia (Approval number 0120-424/2020-3).

### 4.2. Cases

In a period of 1 year, we documented three additional cases of fatal HVA in Slovenia. For the first two cases (cases 1 and 2), an autopsy was performed, and blood and/or bone marrow samples were taken during the autopsy, which was performed less than 24 h after death. Blood was taken from the femoral vein, which was properly prepared and punctured. Venous blood was centrifuged to generate serum for tryptase and sIgE measurement while EDTA-containing whole blood was used for DNA extraction. Bone marrow was harvested from the sternum without fixation. For the third case (case 3), DNA was extracted from an archival blood sample.

### 4.3. Specific IgE and Total Serum Tryptase Testing

The levels of specific IgE to the whole allergen extract of the honeybee, yellow jacket, Polistes, and hornet venom and its recombinant allergens (rApi m 1, rApi m 2, rApi m 3, rApi m 5, rApi m 10, rVes v 1, and rVes v 5) were measured by Immulite 2000Xpi Siemens Healthcare Diagnostics, (Erlangen, Germany) or ImmunoCAP immunoassay (Thermo Fisher Scientific, Uppsala, Sweden). Sensitization was defined as an sIgE level of 0.35 kU/L or more. In patients with a history of HVA and cMCD, sIgE levels were often low; therefore, an sIgE level of 0.10 kU/L or more was also significant [34].

Total serum tryptase (BST) levels were measured using a commercially available fluorescence enzyme ImmunoCAP immunoassay (Thermo Fisher Scientific, Uppsala, Sweden). The lower detection limit was 1 ng/mL, and the normal range for total tryptase levels in serum ranged from 1 to 11.4 ng/mL [35].

### 4.4. KIT p.D816V Missense Variant Assay

Genomic DNA was extracted from 400 µL of EDTA-containing whole blood or bone marrow samples using a QIAamp DNA Mini Kit (Qiagen, Hilden, Germany) according to manufacturer’s instructions. The *KIT* c.2447A > T, p.D816V missense variant, and allele burden were assayed by allele-specific quantitative PCR (qPCR) [12,36] using the ABI 7500 Fast Real-Time PCR system and SDS 2.3 software (Thermo Fisher Scientific).

### 4.5. Tryptase Genotyping

*TPSAB1* and *TPSB2* genotyping was accomplished by droplet digital PCR (ddPCR) as described previously [11,12,37]. *TPSAB1* and *TPSB2* copy numbers were assessed using custom primers and probes specifically targeting α- and β-tryptase sequences together with primers and probes targeting *AP3B1* or *AGO1* as reference genes using a manual droplet generator (Bio-Rad, Hercules, CA, USA), QX200 droplet reader (Bio-Rad), and associated QX Manager software 2.0 (Bio-Rad) [11,12,37].

## 5. Conclusions

In conclusion, reported cases of three individuals with fatal HVA with undiagnosed cMCD, demonstrate that fatal HVA is strongly associated with often unrecognized cMCD. In the first case, anaphylaxis manifested as the initial episode and was fatal; in the other two cases, patients were treated with specific VIT, pointing to the importance of screening all at-risk patients, including those with severe HVA and possibly beekeepers, for proper management and the need for lifelong VIT to prevent unnecessary deaths. Furthermore, patients at the highest risk for fatal HVA, with concomitant cardiovascular and cMCD comorbidities, might not be protected from field stings, even during regular VIT. Therefore, two adrenaline autoinjectors, lifelong VIT, and cotreatment with omalizumab should be considered for high-risk patients to prevent fatal HVA episodes.

## Figures and Tables

**Figure 1 ijms-24-16368-f001:**
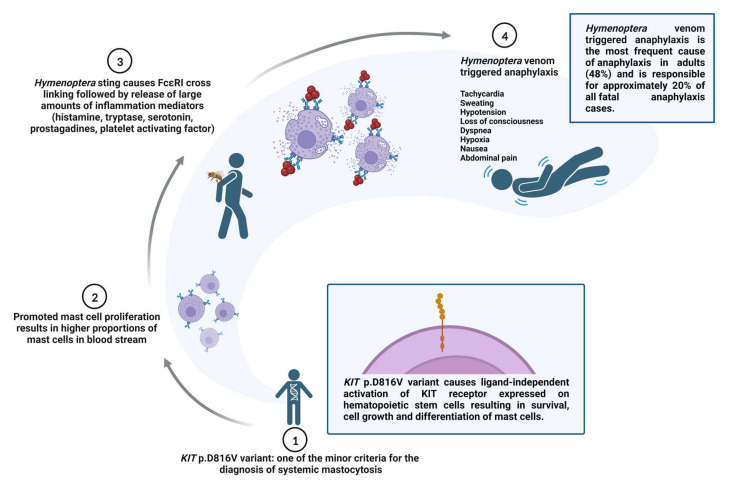
Role of the *KIT* p.D816V variant and clonal mast cell disorders in fatal *Hymenoptera* venom–triggered anaphylaxis. The sequence of events leading to possible fatal anaphylaxis in patients with clonal mast cell disorder is indicated with numbers (1 to 4).

## Data Availability

The data supporting the findings of this study are available from the corresponding author upon reasonable request.

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
