# Peer review of "Fatal Hymenoptera Venom–Triggered Anaphylaxis in Patients with Unrecognized Clonal Mast Cell Disorder—Is Mastocytosis to Blame?"

_ijms, 2023, doi:10.3390/ijms242216368_

Round 1
Reviewer 1 Report
Comments and Suggestions for Authors
I probably expected too much of this paper. Though, the title promissed so much. I, for one, miss a figure, or any graphical content. Probably, the good place for it would be the beggining of the introduction. Placing a figure at ln 47. and presenting a immunological sequejce of ana1pjyoaxis woupd add some value to the introduction.
The introduction as a whole needs a thorough reorganisation. Hymenoptera venom should be introduction to anaphylaxis. With suggested figure added - authors should get an opportunity to better link preclinical with clinical features. The text organization and section/ subsections are totally irrationa and succinct. For instance, section 2. is labelled "results" - what res7lts?
Comments on the Quality of English LanguageTypos ajd grammar!
Reviewer 2 Report
Comments and Suggestions for Authors
The manuscript of Rijavec et al. is a well-written case report of three fatal HVA cases, who – postmortem diagnosed – had an underlying clonal mast cell disease. The authors describe the tragic cases very well and emphasize on the importance of KIT D816V mutational testing in severe HVA cases to take according precautions for these high-risk patients.
Overall, I opt for publication, I just have some minor comments, which should be addressed before that:
1) Lines 15-22: please remove the instructions on how to write the abstract.
2) Line 61: Mastocytosis is a clonal mast cell disease and a mast cell disorder, but not all clonal mast cell diseases and mast cell disorders are mastocytosis. I would suggest to rather write: Clonal mast cell diseases (cMCD), such as mastocytosis and monoclonal mast cell activation syndrome, represent one of the most important…
3) Line 82: I would add “two” to the autoinjectors, just to be clear here as well.
4) Results 2.1 and methods: Here you describe a 10-years analysis of the health register of which 4 people died from yellow jacket and the other three from bees??. How did you get to your three HVA cases? Are they new cases, as your registry analysis happened for the period of 2010-2019, or are thy included in this analysis? Please explain also how you got to the post mortem (bone marrow/blood/serum) samples of these patients and how long post mortem they were taken approximately (because of tryptase levels).
5) Lines 155-157: I would leave out the part of the sentence about Western and Eastern Europe, since Figure 2 of this reference shows that the highest sting related mortality rates are in central Europe (Austria, Slovenia, Hungary) and in northern Europe (Estonia).
6) Lines 220-222: I would add that regularly injecting alum in patients for whom it might not be necessary might bear risks for unwanted side effects. I am open for the author’s opinion on that, but I think, alum should be mentioned in this context.
7) Lines 235-237: This statement is difficult, because cMCDs could still be underdiagnosed, as KIT D816V and tryptase are not standardly measured, but only if there are already clear indications for a cMCD. Now that we know that severe HVA is one of these indication, and all severe HVA cases are screened, the “other half” of non-HVA anaphylactic cMCD patients might be just overlooked. Because they might not know about their disease due to the wide heterogeneity of symptoms and because most doctors apart from allergologists still do not know of/check for this disease. It would therefore be good to also check patients e.g. with early onset osteoporosis for cMCDs to maybe increase the number of “the other half” of cMCD patients, as this is another associated risk factor, which might be independent of HVA.
8) Line 265: In HVA patients with matocytosis, the specific IgE levels are often lower than that. Maybe it would be good to lower the threshold to 0.1kU/L this or next time.
9) Line 303: In these three cases, I am almost certain that you only asked the representatives for informed consent and not the subjects themselves. Please adjust.
Round 2
Reviewer 1 Report
Comments and Suggestions for Authors
IJMSI appreciate all the revisions and I congratulate the authors on their manuscript. However, I do not feel the level of evidence presented is appropriate for IJMS
Comments on the Quality of English LanguagePlease, carefuly proofread!